# Hematopoietic versus Solid Cancers and T Cell Dysfunction: Looking for Similarities and Distinctions

**DOI:** 10.3390/cancers13020284

**Published:** 2021-01-14

**Authors:** Chiara Montironi, Cristina Muñoz-Pinedo, Eric Eldering

**Affiliations:** 1Department of Experimental Immunology, Amsterdam UMC, University of Amsterdam, 1105 AZ Amsterdam, The Netherlands; e.eldering@amsterdamumc.nl; 2Cancer Center Amsterdam, 1105 AZ Amsterdam, The Netherlands; 3Amsterdam Institute for Infection & Immunity, 1105 AZ Amsterdam, The Netherlands; 4Lymphoma and Myeloma Center Amsterdam, 1105 AZ Amsterdam, The Netherlands; 5IDIBELL, Oncobell Program, Cell Death and Metabolism Group, L’Hospitalet, 08908 Barcelona, Spain; cmunoz@idibell.cat

**Keywords:** T cell dysfunction, immunotherapy, metabolism, microenvironment, TME

## Abstract

**Simple Summary:**

Dysfunction of the immune T cell compartment occurs in many hematopoietic as well as solid cancers and hampers successful application of new immunotherapeutic approaches. A complete understanding of T cell dysfunction might improve the outcome of such therapies, but an overview in the various cancers is still lacking. We aim to map areas of similarities and differences in solid versus hematopoietic malignancies, providing a high-level rather than a detailed perspective on T cell dysfunction in those tumors.

**Abstract:**

Cancer cells escape, suppress and exploit the host immune system to sustain themselves, and the tumor microenvironment (TME) actively dampens T cell function by various mechanisms. Over the last years, new immunotherapeutic approaches, such as adoptive chimeric antigen receptor (CAR) T cell therapy and immune checkpoint inhibitors, have been successfully applied for refractory malignancies that could only be treated in a palliative manner previously. Engaging the anti-tumor activity of the immune system, including CAR T cell therapy to target the CD19 B cell antigen, proved to be effective in acute lymphocytic leukemia. In low-grade hematopoietic B cell malignancies, such as chronic lymphocytic leukemia, clinical outcomes have been tempered by cancer-induced T cell dysfunction characterized in part by a state of metabolic lethargy. In multiple myeloma, novel antigens such as BCMA and CD38 are being explored for CAR T cells. In solid cancers, T cell-based immunotherapies have been applied successfully to melanoma and lung cancers, whereas application in e.g., breast cancer lags behind and is modestly effective as yet. The main hurdles for CAR T cell immunotherapy in solid tumors are the lack of suitable antigens, anatomical inaccessibility, and T cell anergy due to immunosuppressive TME. Given the wide range of success and failure of immunotherapies in various cancer types, it is crucial to comprehend the underlying similarities and distinctions in T cell dysfunction. Hence, this review aims at comparing selected, distinct B cell-derived versus solid cancer types and at describing means by which malignant cells and TME might dampen T cell anti-tumor activity, with special focus on immunometabolism. Drawing a meaningful parallel between the efficacy of immunotherapy and the extent of T cell dysfunction will shed light on areas where we can improve immune function to battle cancer.

## 1. Introduction

Roughly a decade ago the interplay between malignant and immune cells has been recognized as an emergent hallmark of cancer [1]. We are already witnessing the advent in the clinic of multiple strategies aimed at addressing this cross-talk and reinforcing immunity in various cancer types. The success of those therapies, generally referred to as ‘immunotherapy’, is dampened by the way cancer cells suppress the immune system, particularly T cells, for their own sustenance. Despite the fact that T cell dysfunction is reported to occur in many hematopoietic and solid cancers, broad mechanistic understanding and overview across the various cancer types is lacking.

The full repertoire of innate and adaptive immune cells plays crucial roles in the pathogenesis of neoplastic diseases, with T cells being considered special players in cancer progression [2] due to their prominent role in destroying pre-malignant cells that could develop into cancer. The onset of an anticancer immune response is a stepwise process known as cancer-immunity cycle [3] (Figure 1), characterized by a series of events that must take place and iteratively self-propagate. Reaching a proper immune response to cancer and tolerance to self-antigens is a crucial balance that has to be maintained in order to prevent autoimmunity. Tolerance is a physiological mechanism in immunity that acts in two separate niches [4], the thymus and the so-called periphery, at different moments during the onset of an immune response. After migrating from the bone marrow to the thymus, the T cells repertoire undergoes the process of central deletion, for the elimination of self-reactive T cells by negative selection. This process of central tolerance is not completely efficient, so additional tolerance is required in the periphery. Peripheral tolerance can be achieved intrinsically via mechanisms regulating the state of T cells (anergy, apoptosis or phenotype skewing) or extrinsically, controlled by the dialogue with other cell types in this second niche, like regulatory T cells (Tregs), dendritic cells (DCs) or myeloid cells [4].

Cancer cells and the tumor niche aid T cell tolerance: immune cells can be subverted to a tolerogenic, inert and dysfunctional state by cancer cells at every level of the cycle, by means of physical interactions and soluble mediators in the immunosuppressive tumor microenvironment (TME), (Figure 1).

Before continuing, in order to better understand how cancer cells impact on each step of the process, it is beneficial to briefly recap what must happen to elicit a successful immune response to cancer. First of all, the genetic aberration(s) leading to oncogenesis generate neoantigens for tumor-specific T cell responses [3]. In order to prime antigen-specific T cells to exert an anticancer response, antigen presentation (signal 1) must be accompanied by other signals: co-stimulation (signal 2) and cytokines (signal 3), (Figure 2). Only if the three signals are delivered, T cells can become activated, start to proliferate and differentiate into effector and memory cells. Following activation, co-stimulatory and co-inhibitory receptors are expressed on the cell surface. The physiological role of inhibitory receptors, also referred to as checkpoints, is to prevent autoimmunity and control the immune response. We propose that these could represent ”signal 4“ during T cell activation, given their role in maintaining T cell response homeostasis. Finally, T cells traffic to the tumor bed and infiltrate the tumor tissue, where activated T cells kill cancer cells [5] via Fas–Fas ligand interaction or by secreting cytotoxic granules containing granzyme and perforin. Various mechanisms can induce tolerance in the TME and limit the effectiveness of the immune response. As will be discussed later, peripheral tolerance is mostly observed in solid tumors, while hematopoietic malignancies show more unique defects in the regulation of central tolerance and priming [6], Table 1.

T cell development and differentiation is shaped and affected by the tumor [7]. Naïve CD4+ (helper) or CD8+ (cytotoxic) lymphocytes differentiate into short lived effector cells upon encountering the antigen. A subset of memory cells remains long term in secondary lymphoid organs or in the tumor bed. It has been widely reported that the individual patient’s T cell profile has an impact on adoptive T cell therapy efficacy [8,9]: chronic antigen exposure and stimulation imposed by cancer leads to the accumulation of terminally differentiated effector T cells, while the memory phenotype is more desirable due to its enhanced persistence and increased activity [10].

As a reflection of the events regulating anticancer immunity, immunotherapy can use at least three sites for intervention: promoting priming, shifting the balance to anti-tumor T cell responses, and counteracting immunosuppression in the TME (Table 2). Immunotherapeutic strategies are broadly classified on the basis of the stage in the cycle where the intervention is attempted: vaccination, adoptive transfer of immune effectors, such as bispecific antibodies or tumor specific T cells with memory phenotype, and immunomodulatory therapy with either checkpoint blockade or cytokines [11,12].

Vaccination strategies will not be discussed here as we focus on T cells, except to state that we are currently witnessing a revival of cancer vaccines, especially due to increasing knowledge of neoantigens [13]. While in the past the approach did not seem to be impactful on cancer treatment, preclinical data are nowadays showing promising results in combining vaccines and immune checkpoint therapy [14], especially when no pre-existing tumour immunity is present. Adoptive T cell therapy requires the isolation of tumor reactive lymphocytes from the patients, their expansion ex vivo in the presence of growth factors, and their re-infusion into patients. Another approach to generate tumor-specific T cells utilizes patient-derived lymphocytes engineered to express a chimeric antigen receptor (CAR) through which they are redirected to recognize the tumor. CARs are chimeric constructs consisting of an antibody region fused to a T cell receptor (TCR) signaling domain, with additional co-stimulatory domains providing signal 2. This results in T cells that are activated and proliferate in vivo upon contact with their antigen but bypassing the need for antigen processing and MHC restriction, since CAR T cells recognize the intact surface antigen. Lastly, therapeutic blockade of checkpoint pathways, administering immune checkpoint inhibitors (ICIs), aims at “removing the breaks” [15] imposed by the expression of inhibitory receptors on T cells.

In general, the mechanisms of resistance to T cell based immunotherapy are overlapping with those used at first by cancers to evade immune destruction and initiate tumor progression [16]. As a consequence, a deep understanding of cancer-induced T cell dysfunction among different cancer types might be the key to more effective therapeutic strategies. Identification of mechanisms associated with cancer-induced T cell dysfunction is crucial to understand predictive prognostic factors for therapy, improve current ICIs regimens using combinatorial approaches and generating CAR T cells that can persist and effectively eliminate tumor cells.

In order to explore T cell dysfunction in cancer, we will address first the efficacy of T cell-based therapies, such as checkpoint blockade and transfer of cellular immune effectors into patients. We aim at mapping areas of similarities and differences in solid and hematopoietic tumors, with the premise that immunotherapy efficacy depends on the state of T cells in cancer patients. Here, T lymphocytes are characterized by features of cancer-induced anergy, senescence and/or exhaustion. Anergy is caused by insufficient co-stimulatory signals during priming; senescence is growth arrest after extensive proliferation; exhaustion is characterized by expression of inhibitory signals after an excessive activating signal due to chronic antigen stimulation [17]. Reversing those features is the key to reach immune reinvigoration towards long-lived T cells that can quickly proliferate and renew themselves upon antigen re-encounter, such as stem cell-like memory T cell precursors.

## 2. Immunotherapy for Solid Tumors

The success of T cell based immunotherapy in solid tumors is limited to checkpoint blockade therapies, with adoptive cell therapy lagging behind [18,19]. Anticancer strategies targeting CTLA-4, PD-1, PD-L1 demonstrated clinical efficacy in many cancer types [20] leading to several FDA-approved antibodies for treating melanoma [21,22], lung carcinoma (mainly, non-squamous non-small-cell lung carcinoma, NSCLC) [23] and DNA-mismatch repair-deficient cancers [24,25]. Despite its success, therapy using ICIs also encounters resistance with mechanisms that resemble those adopted by cancer cells during the initial escape phase of immunoediting [16], ultimately leading to tumor progression.

Loss of immunogenicity is the first mechanism of escape that cancer cells adopt against an immune response. In general, a higher number of identified neoantigens together with more CD8+ tumor infiltrating lymphocytes (TILs) correlate with increased patient survival [26]. Since neoantigens directly reflect the genomic instability of the cancer [16], tumor mutational burden (TMB) has been measured across different cancer types. Tumors with higher TMB (melanoma, lung cancer) are more likely expressing neoantigens recognizable by T cells and more likely responsive to ICIs. Nevertheless, tumor reactive T cells can be also present in cancers with relatively low TMB, such as breast cancer, which suggest that progressing malignant clones are able to escape immunity despite the presence of TILs.

Loss of immunogenicity has obvious negative consequences for adoptive CAR T cell therapy as well: the search for tumor specific antigens is still ongoing in solid tumors. Unlike B cell malignancies in which the tumor cells express the B-cell marker CD19, solid tumors rarely express one tumor specific antigen [27]. They are rather enriched for tumor associated antigens like mesothelin, MUC1 in melanoma, HER2 in breast cancer or express neoantigens, aberrant protein translated by genes with somatic mutations, or proteins which are abnormally expressed in cancer cells (MAGE family, melanoma associated protein). The obvious adverse effect of an antigen co-expressed in non-malignant cells is the off-target effect toxicity and efforts are being made to improve safety, such as co-expressing suicide genes in the CAR construct [28]. Moreover, only a subset of tumor cells might express the antigen, and when it is uniformly expressed, antigen escape might occur: cancers hijack immune recognition through selective outgrowth of cells able to lose immunogenic antigens. Tandem CARs have been generated to address antigen heterogeneity and antigen loss, such as a dual HER2-MUC1 construct that showed promising results in an in vitro model of breast cancer [29].

Another hurdle faced by immunotherapy in solid cancers is tumor infiltration. Physical barriers can prevent TILs from infiltrating the TME [30] and this has a negative impact on the efficacy of ICIs. Stroma and abnormal immature vessels at the core of the tumor have an impact on TILs infiltration, and they act as a barrier for an infused CAR T cellular product [27]. For this reason, an opportunity to enhance CAR T cell efficacy is represented by “armored’’ CAR T constructs, expressing homing receptors or chemokine, such as the anti-mesothelin CAR T construct constitutively expressing the cytokine IL-7 and the chemokine CCL19 (Figure 1) to guide the infused product to the tumor site. This construct showed complete tumor regression in a solid tumor mouse model [31].

When the cancer is immunogenic and T lymphocytes manage to infiltrate the tumor, ICIs seem still the most promising therapy and it is achieving remission in patients that a few years ago were deemed as incurable. Indeed, studies reported that in solid tumors ICIs are more efficient in patients showing TIL infiltrates [32]. An intriguing finding that requires further study is that the number of circulating CD8+ T cells is negatively associated with the durable effect of ICIs in NSCLC [33], possibly reflecting tumor T cell infiltration. However, our knowledge of the role of pre-existing immunity in the efficacy of checkpoint blockade is still very limited.

Adoptive immunotherapy using TILs or CAR T cells seems to require further improvements, for several reasons. Tumor infiltrating lymphocytes are dysfunctional and exhausted, due to chronic stimulation and their coping mechanisms to survive the microenvironment at the tumor site. Thus, the obvious limitation to TIL adoptive cell therapy is the expansion of sufficient numbers of reactive T cells in vitro, because TILs might be already terminally differentiated due to chronic stimulation in the TME. T cell phenotypes within the tumor do not mirror the immune composition in peripheral blood [34], where T cells do not show such an exhausted phenotype, as exemplified by a study conducted in melanoma-patients where the ratio and absolute numbers of CD4+ and CD8+ T cells in peripheral blood were found comparable to age-matched healthy donors [35], as well as the frequencies of memory T cell subsets, which did not show increased expression of inhibitory receptors. Those findings seem promising for the generation of CAR T cells for patients with solid tumors since peripheral lymphocytes seem better equipped to attack cancer cells. However, it is important to take into account that these peripheral T cells might not express the right homing receptors to reach the tumor, so they should be “armored” as previously indicated. A second challenge is represented by the higher proportion of Tregs found in the peripheral blood of solid malignancies, including melanoma, lung and breast cancer [35,36] that may negatively influence CAR T cell composition [37].

Even when the generation of a CAR T cell infusion product is successful and able to reach the tumor site, those cells have to face the TME. The TME is inhabited by stromal cells and suppressive immune cells including myeloid-derived suppressor cells (MDSCs), tumor associated macrophages with M2 phenotype and Tregs. Together with malignant cells, those cell types express ligands for inhibitory receptors on T cells and secrete soluble factors impacting T cell priming, survival and function, which we will discuss later. Microenvironmental clues also matter in hematopoietic tumors.

## 3. Immunotherapy in Hematopoietic Malignancies

Hematopoietic malignancies develop in the same niche in which immune responses are generated, suggesting that these tumors fail to trigger immune sensing mechanisms, or effectively impair anti-tumor immune responses when they do take place [6]. T cell priming usually spontaneously occurs at least in a subset of solid cancers and it is then followed by functional impairment at the tumor site. T cell tolerance in hematological malignancies is regulated at the central level, where tolerogenic antigen-presenting cells lead to a defective priming upon initial antigen encounter. Ineffective priming could be one of the reasons why ICI are less efficient in hematopoietic malignancies when compared to the successes in solid tumors, since the lack of a proper immune response does not derive from checkpoints preventing the killing of cancer cells, but rather by a state of immunological ignorance. Moreover, later on we will discuss whether T cells in multiple myeloma (MM), for example, exhibit the features of senescence or anergy, rather than exhaustion [38,39,40]. This distinction could provide a second reason for the failure of ICIs in hematopoietic malignancies. ICIs showed some success in Hodgkin lymphoma [41] using the anti-PD-1 nivolumab, but have to make an impact yet in other hematopoietic cancers such as chronic lymphocytic leukemia (CLL), MM or diffuse large B-cell lymphoma (DLBCL) [42]. Interestingly, the classical Hodgkin lymphoma is characterized by genetic alterations resulting in constitutive expression of PD-1 ligands [43], explaining the reason of this success. In other hematopoietic malignancies, trials are still ongoing, testing combinatorial approaches [5] with either conventional cytotoxic agents or monoclonal antibodies (mAbs) such as the anti-CD20 rituximab [44].

To date, the major successes of CAR T cell therapies have been recorded in hematological tumors [45,46,47]. The first FDA approvals were witnessed in 2017, with Kymriah^®^ (Novartis, Basel, Switzerland) and Yescarta^®^ (Kite Pharma, Santa Monica, CA, USA) targeting the B cell antigen CD19 to treat B-acute lymphocytic leukemia (B-ALL) and DLBCL, respectively. CAR T cell products showed high efficacy in high-grade malignancies, but not in low-grade hematopoietic cancers. The pathogenesis of those two categories of tumors is different: high-grade tumors, such as B-ALL, are fast growing and characterized by immature precursors, while low-grade tumors, like CLL, show a slower growth of mature B cells. This subtle and persistent growth induces chronic stimulation, a state of cancer-induced T cell dysfunction and acquisition of a terminally differentiated phenotype. T cell defects in CLL due to disease and/or therapy impair ex vivo expansion and response to CAR T cells. Previous therapies for CLL, including alkylators and fludarabine, have a profound negative effect on T cell function, exacerbating the T cell defect in CLL. The Bruton tyrosine kinase (BTK) inhibitor Ibrutinib, instead, may potentially improve T cell immunity [48] and may be administered in combination with CAR T cell therapy or ICIs [49]. A recent trial using lymphodepletion and CD19 CAR T cells showed very promising results in CLL-patients after failure of Ibrutinib treatment [50], most likely because of the effect of this drug on the T cell compartment. Concerning the combinatorial approach of Ibrutinib with ICIs such as Nivolumab, an early phase trial [49] showed that the combination had no additional efficacy compared to single agents therapies in patients with relapsed/refractory CLL and other hematopoietic malignancies.

In order to compare the hurdles of adoptive T cell therapy in hematopoietic cancers with the ones described for solid tumors, we will first discuss antigen selection and immunosuppressive nature of TME. While finding an optimal CAR T cell target is still a challenge in solid cancers, CD19 appears as a highly suitable target in B cell hematologic malignancies for several reasons: its expression is ubiquitous on malignant and normal B cells. Loss of normal B cells can be overcome by immunoglobulin replacement therapy [37]. A more serious concern is neurotoxicity associated with CD19 targeting, most likely deriving from CD19 expression on mural cells pivotal in maintaining the integrity of brain-blood barrier [51]. This off-target effect makes CD19 a less ideal target than previously expected. Moreover, as previously discussed, a second downside is that antigen escape, widely accountable for CAR-T cell resistance, has been reported with CD19 CAR as well, in B-ALL [52] and other B cell malignancies [53,54]. Therefore, other targets are currently being explored, such as B-cell maturation antigen (BCMA) in MM [55] or CD22 in B-ALL [56]. The approach of dual or tandem CAR has been pursued for hematopoietic cancers as well, and some constructs targeting CD19/20 or CD19/CD22 entered clinical trials in hematological malignancies [57].

Regarding the immunosuppressive feature of the tumor niche, hematopoietic cancers can be compared to solid tumors when considering that the solid component is still significant in leukemia, lymphoma or myeloma, where secondary or primary lymphoid organs function as the TME. MM is homing in the bone marrow and like in solid tumors, T cells at the tumor site are more severely impaired than the ones in peripheral blood [38]. Those compartments are inhabited by immune cells, whose function, rather than eliminating leukemic cells, is redirected to promote cancer cells survival and to generate an immunosuppressive and protumorigenic microenvironment [58,59]. Profound changes in the immune compartment are reported to occur in all hematopoietic cancer types, due to the fact that malignant cells themselves may be part of the immune reaction in different ways and can alter the rest of the players in adaptive immunity. For this reason, it might be speculated that tumor-host interactions in B cell tumors play a special role in tumor initiation and progression. Moreover, cancers like MM or lymphoma are homing in niches in which T cell development take place. Together, those findings suggest indeed that the immune compartment is dysfunctional since the onset of the neoplastic disease in hematopoietic malignancies, raising the “chicken–egg” dilemma: which comes first, cancer initiation or immune dysfunction?

Compared to solid cancers, hematopoietic malignancies show more profound abnormalities in T cell development and differentiation, which are affecting also the blood compartment. Multiple myeloma patients show quantitative and functional T cell abnormalities [60], with a decrease in the CD4+/CD8+ ratio [61] associated with disease progression [62]. T cell subsets skewed in favor of an immunosuppressive state [63], as shown by an increase ratio of Tregs when compared to the inflammatory T helper 17 (Th17) subset. Data suggests that the acquired T cell dysfunction derives from the specific interaction of myeloma cells with T cells resulting in impaired T cell immunity against the tumor, but the T cell response against external antigens is mostly not affected [64]. Clonally expanded T cells are present in the blood of myeloma patients, with an incidence between 50–75% [40]. Those clones are functionally resembling more a senescence phenotype rather than exhaustion [40], which may also explain why checkpoint blockade using PD-1 or CTLA-4 inhibitors was not successful in trials. In a study of long-term survivors of MM, those clonally expanded T cells show regained proliferative capacity, suggesting that this senescent-like state is reversible. Moreover, survivors also showed a normal or reduced Treg/Th17 ratio [65].

Immune dysregulation is also prominent feature of CLL from its early stage and alterations within the T cell compartment become more apparent with disease progression or after treatment [66]. The absolute CD4+ and CD8+ counts are increased, with CD8+ subset showing a higher relative increase which accounts for the reduction of the CD4+/CD8+ ratio [67,68,69]. It has been speculated that the expansion of T cell counts would represent an attempted specific immune response against CLL [66], yet CLL-specific T cells have not been reported so far. Together with quantitative changes, CD8+ and CD4+ T cells also show functional defects [70], with a marked alteration in helper activity [71]. Another factor likely contributing to immune suppression is the presence of an increased proportion of Tregs [72].

There are few studies regarding the impact of T cell dysfunction in B-ALL development and relapse after (CAR T cell) therapy, and these focused mostly on the bone marrow microenvironment [73,74], which is where this malignancy of immature progenitor B cell blasts initially takes place. It has been shown that B-ALL is associated with loss of CD8+ T cells, as indicated by a significantly lower frequencies of CD8+ T cells in the patients’ bone marrow compared to healthy controls, together with accumulation of late stage effector CD4+ T cells [73]. These data indicated that the composition of bone marrow T cells is profoundly affected by leukemic blasts, provoking a late-stage differentiation. Studies on peripheral blood of B-ALL patients showed again an accumulation of the Treg subset as compared to healthy donors [75]. As previously stated, an increased frequency of Tregs in peripheral blood is also a common feature of solid tumors, indicating that the presence of this immunosuppressive subset might play a common pivotal role in cancer progression. In the next sections we will gradually abandon the separation between solid and hematopoietic cancers and discuss more deeply contact-dependent interactions, as well as soluble factors by which tumors cells and/or other bystander cells are affecting T cell development and functions in the context of cancer-immunity-cycle (Figure 1).

## 4. The Immunosuppressive Microenvironment: Physical Interactions

As hypothesized by Curran et al. [6], tumor specific T cells are primed but become functionally impaired at the tumor site in solid tumors. By contrast in leukemia or lymphoma T cells are not properly activated, but rather energized upon antigen encounter. This notion is supported by data collected from a murine model of acute leukemia [76], in which the implantation of leukemic cells in situ to mimic solid tumor induced an antigen-specific CD8+ T cell response while the systemic engraftment of leukemic cells induced a T cell tolerogenic state, characterized by defect in proliferation. The tolerant phenotype could be abrogated when administrating an agonistic anti-CD40 antibody, suggesting a defect on antigen presentation and antigen presenting cell (APC) activation as drivers of the tolerogenic state [76]. This approach was actually implemented in B-ALL-patients after allogeneic stem cell transplantation, where it was able to induce immune responses reactive against leukemic blasts [77]. Other indications of the feasibility of targeting CD40-CD40L interactions come from a phase I study conducted in CLL patients [78], where preliminary clinical responses such as the reduction of leukemic cell counts were observed. Defects in priming are also observed in solid cancers with a low mutational burden and the use of CD40 agonists is able to potentiate T cell responses to solid tumors and might be used in combination to improve ICIs therapy [79,80].

Co-stimulation and co-inhibition is another process in T cell activation which is often impaired in cancer. As previously discussed, the expression of inhibitory receptors (immune checkpoints, “signal 4”) on T cells represents a physiological mechanism aimed at mitigating tissue damage and autoimmunity [81] (Figure 2). This mechanism is exploited by tumors: cancers suppress immune responses by expressing ligands for negative regulatory receptors on T cells, such as CTLA-4, PD-1, LAG-3 and TIM-3. CTLA-4 signaling is more involved in preventing the initiation of a T cell response in the lymph nodes, while PD-1 serves to limit T cell activity in the TME [82]. After TCR engagement, during the priming phase, CTLA-4 is upregulated to attenuate T cell responses and limit the expansion of autoreactive T cells. After the initial success in melanoma, anti-CTLA-4 antibodies like ipilimumab were tested in solid tumors such as breast cancer with limited efficacy [83], though it has been recently approved for the treatment of lung cancer in combination with the anti-PD1 antibody nivolumab in advanced NSCLC [84].

Another molecular mechanism subverted by cancer cells is the very last step of the cancer-immunity cycle, namely the killing of tumor cells by T cells. A functional CRISPR/Cas9-based genome-wide knockout screen performed in B-ALL identified the death receptor signaling pathway as a resistance mechanism to CD19 CAR T-induced cell death [85], suggesting that cancer can also escape CAR T cell killing. Lacking the pro-apoptotic signaling molecules BH3-interacting domain death agonist (BID) or fas-associated protein with death domain (FADD), B-ALL cells were resistant to CD19 CAR T cytotoxicity, leading to disease progression in mice, persistence of tumor cells, which exacerbated T cell dysfunction. Those findings were validated ex vivo after CD19 CAR T cell treatment in B-ALL patients [85]: expression of death receptor pathway genes in pre-treatment samples correlated with CAR T cell expansion and persistence, as well as patient overall survival.

By-stander cells in TME can be subverted by cancer cells to suppress T cell function: it is the case of immunosuppressive M2 macrophages, Tregs and MDSCs. The increased fraction of Tregs identified in both solid and hematopoietic cancers inhibits anti-tumor immunity by various mechanisms, such as the constitutive expression of CTLA-4, inhibiting CD80/86 co-stimulatory signals on APC (Figure 2) or direct killing of effector T cells via Fas-FasL [86]. MDSCs are also increased in both solid (e.g., breast cancer and melanoma) and hematopoietic malignancies (e.g., MM and CLL) [87,88,89,90,91]. They have protumorigenic functions in the TME via multiple mechanisms [92]: induction of immunosuppressive cells, such as Tregs [93] or differentiation of macrophages into the M2 phenotype, blocking of lymphocyte homing via downregulation of cell adhesion molecules on T cells, expression of checkpoints such as PD-L1 [94]. M2 macrophages are often expressing PD-L1 or ligands for CTLA-4 [95], providing T cells with the inhibitory “signal 4” dampening activation.

For reason of limited space, we indicated excellent reviews on bystander cells, extensively discussing the role of Tregs [86], macrophages [95] and MDSCs [87,91,92]. Their immunosuppressive role is gaining importance as it might turn out to be one of the main reasons for immunotherapy failure. Another mechanism by which by-stander cells contribute to the generation of an immunosuppressive and protumorigenic milieu is the secretion of immunosuppressive cytokines [96] and other soluble signals, which will be discussed in the next section.

## 5. The Immunosuppressive Microenvironment: Soluble Signals

In the context of the cancer-immunity cycle, soluble signals such as cytokines represent signal 3 to balance TCR signaling either to a potent immune response (IL-2, IL-17) or a tolerogenic state (IL-10, IL-6), (Figure 1). Other than cytokines, soluble signals that mediate cell-cell communication in the TME are also represented by exosomes [97,98] and (immunomodulatory) metabolites.

Inflammatory cytokines such as TNF-α, IL-6, TGF-β, and IL-10, have been shown to participate in cancer initiation and progression [99], depending on the balance of pro- and anti-inflammatory cytokines and their relative abundance. Besides cancer cells, immune cells, including M2 macrophages, MDSCs and Tregs, and stromal cells, such as fibroblasts and endothelial cells, synthesize them to regulate proliferation, cell survival, differentiation, immune cell activation, cell migration, and death [96]. The most known cytokines to induce T cell dysfunction are IL-10, IL-6 and TNF-α, while TGF-β role in cancer is mostly related to epithelial-to-mesenchymal transition (EMT), invasion and metastasis in melanoma [100], breast [101,102] and lung cancer [103].

IL-10 is a potent anti-inflammatory cytokine, secreted by cancer and immune cells [104,105]. IL-10 acts on an autocrine manner on naïve T cells promoting their differentiation into Tregs and on effector T cells limiting their proliferation [106]. Moreover, IL-10 suppresses APC pro-inflammatory responses: due to its immunosuppressive effect on DCs and macrophages, IL-10 can dampen antigen presentation allowing tumor cells to evade immune surveillance [107] (Figure 1). IL-10 plays a role in inducing suppression of anti-tumor immunity in CLL [108], contributing to the state of cancer-induced T cell dysfunction.

Another pro-inflammatory cytokine with a typical pro-tumorigenic and immunosuppressive effect is IL-6. Elevated IL-6 levels have been detected in serum of patients with systemic cancers. IL-6 has a role in multiple myeloma development, where malignant cells induces senescence in bone marrow stromal cells [39] and cytotoxic clonal T cells [40]. Present at high levels in MM microenvironment, IL-6 plays a crucial role in preventing apoptosis in favor of senescence [109].

Cytokines and soluble factors in general play a relevant role in shaping the immune composition in solid cancer as well. To give an example, CD8+ T cells infiltrating the lung TME and the pleural effusion are dysfunctional: they are unresponsive or poorly responsive to any T cell activating stimulus and functionally impaired [110]. Surprisingly, TILs or T cells in the pleural effusion show an elevated ratio of memory CD8+ T cells, attracted in a subset-specific manner by chemokines such as CCL21, CCL5, CCL2 [111] (Figure 1). Effector CD8+ T cells are less represented in the TIL population. Two mechanisms could lead to the absence of this subset, namely cell death or tumor cells/TME blocking the differentiation process from memory cells to terminally differentiated effector CD8+ T cells. Immunosuppressive factors in the microenvironment can prevent such differentiation. Lung cancer cell lines secrete IL-6, IL-10 and TGFβ [112,113], which are also found in pleural effusions [103]. IL-8 is also secreted by lung cancer cells: this factor is a potent chemoattractant (Figure 1) and recent studies indicated that, besides being involved in EMT of epithelial cancer cells, paracrine signaling by tumor derived IL-8 promotes recruiting of MDSCs into the tumor [114], dampening anti-tumor responses.

By-stander cells in the TME are also educated by cancer cells to secrete soluble factors negatively affecting T cell immunity. Besides the above mentioned cytokines, MDSCs and M2 macrophages release nitric oxide (NO) and reactive oxygen species (ROS) [91] that impair T cell activation, via disruption of TCR and eventually induce apoptosis of T lymphocytes [115]. Another mechanism adopted by MDSCs is the expression of the enzyme indoleamine 2,3-dioxygenase (IDO), involved in the catabolism of the metabolite tryptophan that has profound immunosuppressive effects on T cells.

The tumor microenvironment is poor in nutrients due to the high metabolic demand of malignant cells; solid cancers are characterized by gradients of nutrients, hypoxia and by-products which can act as immunosuppressive metabolites. To give some examples, adenosine is known to affect priming [116] and function [117] of T lymphocytes; kynurenine mediates exhaustion of T cells in melanoma [118], lactate can impair effector functions such as IFNγ production [119] (Figure 1).

We will expand in the next section on why metabolism is important in cancer immunity (dys)function.

## 6. Immunotherapy Efficacy Is Linked to T Cell Fitness—The Survival of the Fittest

Metabolic reprogramming is inherently linked to T cell development, activation, function, differentiation, and survival [120,121]. After T cell activation, increased demands for energy (ATP), as well as biosynthetic precursors for proliferation, are met by rewiring cellular metabolism. Upon stimulation, co-stimulatory molecules such as CD28 or 4-1BB allow previously quiescent T cells to augment their glycolytic capacity. Downstream to TCR engagement and co-stimulation, activation of PI3K, Akt, and mTOR triggers the switch to anabolic metabolism via the transcription factors Myc and hypoxia-inducible factor 1 (HIF1) [122,123]. Interestingly, inhibitory receptors such as PD-1 and CTLA-4 partially act by reducing activation-induced glucose uptake [122,124]. Subsequent to glycolysis, oxidative metabolism is augmented, together with increase in the mitochondrial mass (Figure 3). Respiration is therefore also essential for proliferating T cells.

Glycolysis is required to obtain energy, biomass and precursors essential to initiate the differentiation program towards dividing cells and is also crucial for the production of effector molecules like IFN-γ, IL-17 and Granzyme B in T cells [125,126,127,128].

Rewiring of cellular metabolism is also a well-known feature of cancer cells [1], which are adopting a so-called ”aerobic glycolysis”, otherwise referred to as Warburg effect. Even in presence of oxygen, cancer cells prefer the fermentation of glucose into lactate over oxidative metabolism. This choice does not depend on mitochondrial defects nor on impaired oxidative phosphorylation, which is instead required for cancer progression [129]. Glycolysis has various benefits [130] for rapidly proliferating cancer cells, counteracting the selective pressures imposed by TME and, perhaps, by the immune system. At the same time, cancer cells maintain a high metabolic plasticity [131] in order to cope with fluctuations in nutrients and oxygen levels during cancer progression, resulting in the survival of the fittest. Additionally, in multiple tumors, including breast and colon cancer, a fraction of quiescent cells presumably maintaining an oxidative phosphorylation (OXPHOS)-based metabolism [132] is responsible for resistance and relapse to therapy, as well as immune escape [133].

The Warburg effect indeed describes tumor metabolism only partially [134], and the high metabolic heterogeneity within tumors is achieved through a complex network of interactions between different compartments, such as cancer cells, by-stander cells and stromal cells. This metabolic coupling allows the transfer of metabolites from stromal cells to support cancer cells in their response to fluctuations of nutrients [135]. Interestingly, a recent model, namely the reverse Warburg effect, describes a two-compartment interplay in which stromal and by-stander cells are induced by cancer to assume a glycolytic phenotype, whose catabolic products (lactate, pyruvate, ketone bodies) are then used by cancer cells for mitochondrial OXPHOS. It has been recently proposed that a more complex two-dimensional model including stroma and cancer metabolism should be taken into account when assessing immunotherapy efficacy [136].

Studies correlating cancer metabolism and immune dysfunction are flourishing. Interestingly, cytokine secretion by tumor cells is regulated by external stimuli and stressors such as glucose deprivation, as shown in a model of lung cancer [137] and this might represent an additional mechanism of immune regulation in solid tumors.

As shown by experiments conducted in murine models of melanoma and sarcoma, in solid tumors the mechanisms underlying T cell metabolic dysfunction can be rooted in metabolic competition for glucose [119,138], or in the immunosuppressive effect of lactate [139] or other by-products of cancer cells, such as the previously mentioned tryptophan or kynurenine (Figure 1).

In addition to melanoma and sarcoma murine models, in CLL patients T cells have defects in activation, accompanied by the inability to properly upregulate glucose uptake and glycolytic metabolism [140]. Immune suppression by glucose competition has apparently been excluded in CLL, as well as a role for lactate in immune suppression [141]. CLL seems indeed an exception to the Warburg rule: CLL cells take advantage from catabolism of fatty acids, while maintaining a high metabolic plasticity to counteract selective pressures in TME [142]. T cells from B-ALL patient showed a similar deficiency in activation and glycolytic metabolism upon stimulation [140]. Those defects did not impede the generation of functional and highly persistent CAR T cells which lead to enormous successes in B-ALL treatment, while the efficacy is still very limited in CLL. It might be speculated that T cell dysfunction in CLL is exacerbated by chronic stimulation. In a recent report, it has been shown in an in vitro model that persistent antigenic stimulation leads to a loss of mitochondrial function [143], due to the induction of mitochondrial oxidative stress that limits ATP production via OXPHOS. Other studies are underscoring the importance of mitochondria for T cell function [144], such as a recent report showing that T cell exhaustion is reinforced by deregulated mitochondrial dynamics (fission and fusion) regulating the turnover of those organelles [145]. Interestingly, the ultrastructure of mitochondria also differs in memory versus effector T cells [146]. The latter are characterized by punctate mitochondria, while memory T cells maintain a highly fused network of these organelles (Figure 2).

In addition, metabolic plasticity also regulates differentiation and skewing of T cells, with terminally exhausted effector cells showing a glycolytic metabolism, in contrast to fatty acid β-oxidation characterizing memory subsets [147], (Figure 3). Thus, the hypoxic, acidic, scarce-in-nutrients tumor microenvironment has a negative impact on T cell differentiation. The hypoxic conditions in the TME, for example, prevent the acquisition of the memory T cell phenotype, which relies heavily on oxygen [27]. It is widely reported that the memory phenotype increases T cell survival, proliferation and prolonged presence of TILs and CAR T cells at tumor sites [143,148]. Patients with prevalence of memory T cells at the tumor site have increased survival [149] and the state of differentiation exhibited by the apheresed T cells affects the efficacy and persistence of the infusion CAR T cell product [150].

In the last decade we witnessed the rise of the immunometabolism field and studies on the topic have increased exponentially. Reports describing the alterations in T cell metabolism can have profound implications in the optimization of immunotherapeutic strategies. In view of those findings, researchers are making efforts in the direction of manipulating the metabolism of either TILs or CAR T cells to more efficiently utilize nutrients, with the aim of maximizing their performance and prolong their survival, eventually improving the efficacy of cancer immunotherapy. The design of CAR constructs can be improved by ‘armoring’ T cells with enhanced metabolic functions. On the other hand, this knowledge could also be beneficial in improving ICIs therapy when combined with drugs targeting and fine-tuning metabolism as adjuvant therapy.

## 7. Conclusions

In summary, hematopoietic and solid cancers are showing different defect in central versus peripheral tolerance, with B-cell lineage malignancies displaying a more unique defect in TCR selection and priming, and solid cancers presenting adequate immune response initiation which is then dampened at the tumor site.

Many trials are testing combinatorial approaches of immunotherapy with other treatments. A strategy could be optimizing the schedule of immunotherapy administration, for example after surgery to remove tumor cells inducing T cell dysfunction, taking care not to exacerbate it via chemotherapy. An early use of immunotherapy to treat patients with early-stage cancers could lead to an improved efficacy. To date, immunotherapy has primarily been used in patients with advanced stage or relapsed cancers. However, multiple trials are addressing the benefit of the use of immunotherapy in early-stage tumors. Hopefully, those ongoing trials will reveal even better results than those obtained in advanced settings.

Studies reporting the importance of metabolism in cancer immunotherapy are flourishing, but still a complete overview of metabolic interactions between malignant and immune cells in the various cancer types is lacking. A deeper investigation of immunometabolism should indeed lead to a profound improvement of immunotherapeutic strategies: ICIs approaches could benefit from the administration of adjuvant therapies targeting metabolism and nutrient utilization, and CAR T cell persistence could be boosted by armoring the chimeric TCR with genes involved in the regulation of cellular metabolism.

## Figures and Tables

**Figure 1 cancers-13-00284-f001:**
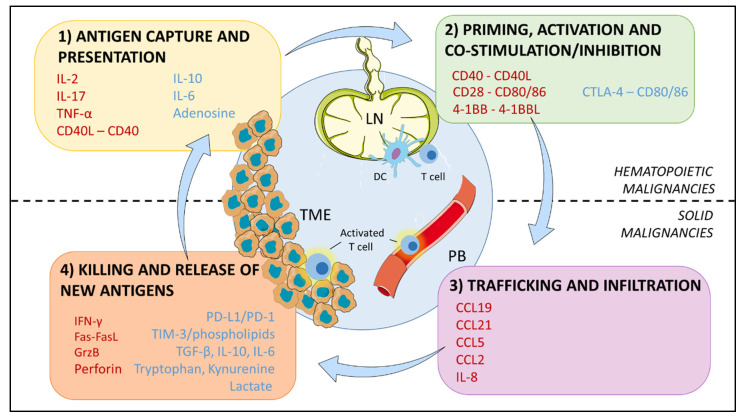
Cancer-immunity cycle. The immune response against cancer starts in the tumor microenvironment (TME) where antigens released by cancer cells are captured by dendritic cells (DCs), followed by antigen presentation to T cells in the lymph node (LN) for their priming. Activated T cells traffic through peripheral blood (PB) and infiltrate the tumor bed to kill cancer cells. The cycle is propagating with new antigen release. Immunostimulatory and inhibitory factors promoting or suppressing the cycle are indicated in red and blue, respectively. Antigen presentation and priming is the main mechanism affected in B cell malignancies, while T cell dysfunction in solid cancers is predominantly due to defective tumor infiltration and killing in the TME. Figure adapted from REF [3] to match the scope of this review.

**Figure 2 cancers-13-00284-f002:**
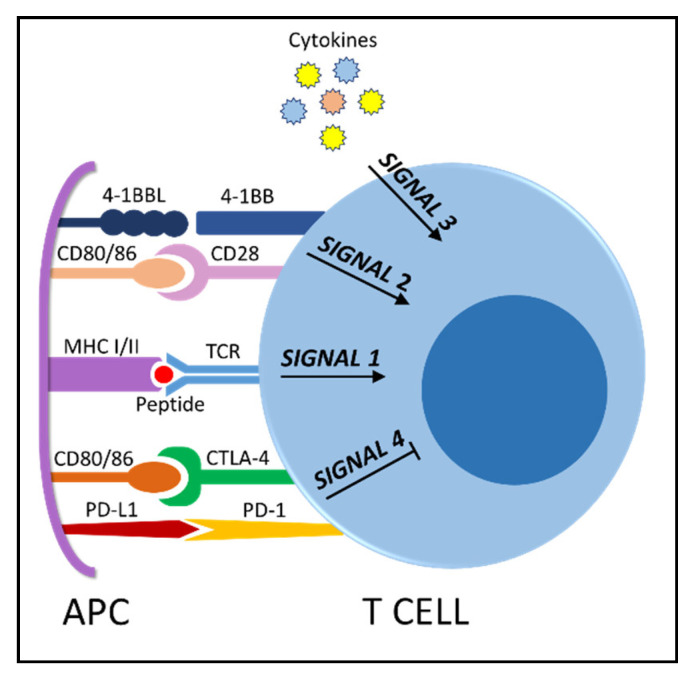
T cell activation requires 3 signals: T cell receptor (TCR) stimulation, co-stimulatory molecules such as CD28 or 4-1BB, immunostimulatory cytokines. Activating signals can be counteracted by inhibitory molecules (checkpoints, “signal 4”) such as CTLA-4 and PD-1. The respective ligands on the antigen presenting cell (APC) are indicated.

**Figure 3 cancers-13-00284-f003:**
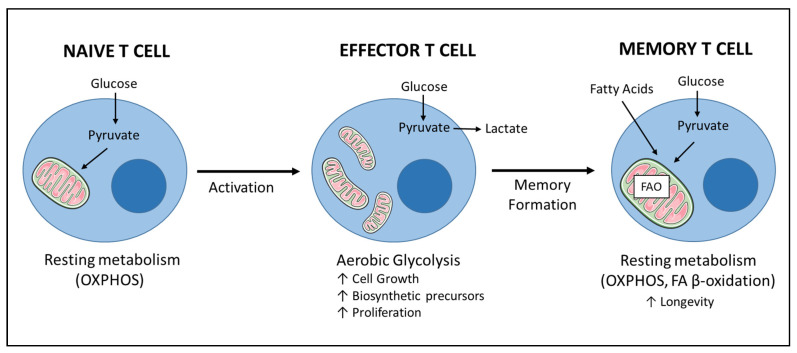
Simplified representation of metabolic remodeling during T cell activation and acquisition of memory phenotype. T lymphocytes exhibit distinct energy demands according to their differentiation status. Naïve, quiescent T cells predominantly use a resting metabolism relying on oxidative phosphorylation (OXPHOS) to produce ATP: the majority of pyruvate derived from glucose and other fuels (not shown) enters the mitochondria. Early after activation, effector T cells switch to a more glycolytic state to sustain cell growth and proliferation: glucose is mainly fermented into lactate; mitochondria appear punctate. After memory formation, T cells acquire a quiescent metabolic state predominantly fueled by β-oxidation of fatty acids (FAO) that ensures longevity after priming. Memory T cells are also characterized by a highly fused mitochondrial network. Note: there is an ongoing debate whether memory T cells derive from effector T cells, and/or directly from naive precursors after T cell activation. For clarity we have represented the first option, while not excluding the latter.

**Table 1 cancers-13-00284-t001:** Similarities and distinctions in T cell dysfunction in solid versus hematopoietic cancers.

Similarities	Solid Malignancies	B-Cell Malignancies
T cell dysfunction mechanism	Trafficking/infiltration, Exhaustion in TME	Priming and activation, Anergy, Senescence
Hurdles to adoptive (CAR) T cell therapy	Antigen choice, tumor infiltration	T cell state pre-expansion
Immunosuppression by soluble factors	By cancer cells, cancer associated fibroblasts and innate immune cells	By B cancer cells or by-stander cells
Immunosuppression by physical interactions	Co-inhibition; killing; interaction with by-stander cells	Priming, co-stimulation; killing; interaction with by-stander cells
T cell fitness and metabolism	Metabolic competition restricts T cells	Unknown mechanism(s) alter T cell metabolism

All cancers types induce immune dysfunction, as indicated in the first column, which lists similar aspects of immune defects. The distinctive aspects in solid versus B cell malignancies are summarized in their respective columns and addressed in further detail in the text.

**Table 2 cancers-13-00284-t002:** Broad classification of immunotherapeutic strategies and respective sites of intervention.

Therapy	Intervention in the Cancer- Immunity Cycle	Strategy
Cancer Vaccines	Promoting T cell Priming	Not discussed
Transfer of immune effectors	Enhance or generate T cell response	TILs
		CAR T cells
		bispecific antibodies
Immunomodulation	Counteract immunosuppression	Immune Checkpoint Inhibitors (ICIs)
		Cytokines and cytokine blockade

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
