# Peer review of "Hematopoietic versus Solid Cancers and T Cell Dysfunction: Looking for Similarities and Distinctions"

_cancers, 2021, doi:10.3390/cancers13020284_

Round 1
Reviewer 1 Report
This review gives a nice overview of the immune reaction, with focus on Tcells, to tumors and the pathways and hinderances of immune therapy. In summary this reviewer sees this manuscipt as a systematic and therefore important overview, but several improvements should be made.
Graphic schemes need improvement, since they do not always fit the text and do not include all the implied and wanted information. Certain tables would improve the manuscript (see below).
A graphic depiction of all cell types involved and especially one of summarising the T-cell states would be helpfull.
A listing (table) of the involved molecule, cytokines etc. with regard to their origin/pathway and their pro- or anti-tumor activity would be helpfull, as would be a table of therapeutics described in this manuscript
The refernce list is very long and has several up-to date papers and several older references – yet it can be imporved.
A direct juxtapposition of similarites and differences in addition to the (necessary) separate paragraphes would greatly enhance the manuscript.
Section1 Introduction
In the introduction the focus lies on Tcells, yet severl times ‚ immunesuppression by TME‘ and antigene presentation is mentioned. Here somewhat more details would be helpfull.
Several points of action of immune therapy are discribed systematically. Yet this could be improved a better subdivision in paragraphs (with subheadings) and a table or figure.
Figure 1: The cell types listed in the legend are not to be found in the figure – an entry point is missing. Factors are grouped, but not specified – according to cell type. Figure, figure legend and text do not fit together.
Indeed Figure 1 seems to be a melted down version of the main Figure concept of Reference 3. But information content of Figure 1 is not satisfiing for this manuscript. Therefore this Figure 1 needs reworking. It should be considered, if a more recent review of authors of ref 3 should be cited.
Section2 Immunotherapy for solid tumors:
Immunotherapy is by the authors own description in the Introduction not limited to Tcell transfer, but includes checkpoint therapy. This should be more reflected in this section – or the heading must be adjusted.
A reference for ‚DNA-mismatch repair deficient cancers‘ is missing.
Section3: Immunotherapy for hematopoetic malignancies
He authors correctly make the point, that the tumor cells and immune cells originate in the same compartment. Yet this is going even further: malignant cells can be part of the immune reaction, in different ways, including changing homoestasis.
In line 244 the comparison of solid and hematopoetic tumors/target strategies starts, therefore a new heading is required.
Figure 3 is very simplified. More details would be helpfull. E. g. fatty acid metablolism ismentioned in the text and referred to Figure 3 (where it cannot be found).
Section 6: Survival of the fittest
Discussion of metabolic changes should include more interactions between tumor cells and immune cells, including more details on Warburg and reverse Warburg effects. It is important for Tcells ‚to be fit‘ yet, metabolic interactions are not only competition for nutrition as energy. Here selection of references could be improved.
Reviewer 2 Report
The authors have compiled a solid overview of immune defects induced by solid tumors in comparison and contrast to blood cancers. This is discussed in a well structured and informative manuscript which explains the various aspects of the by the cancer suppressed immune cells and the respective mechanisms of a microenvironment that promotes escape of the various cancers from the immune system. Also implications for immune therapy are discribed. This review seems to meet all criteria to be helpful for interested readers in this topic.
Reviewer 3 Report
In general, the manuscript is well written and provided a comprehensive and critical review about many important points on T cell dysfunction in cancer (solid and hematopoietic) and its influence in CAR T cell and ICI immunotherapies. However, some revisions must be done before publication.
Minor concerns:
General: The first part of the manuscript reflects very well its title, however after read the complete work, one is left with the impression that the title does not include all the concepts reviewed. Please I suggest aim this by changing the title or put more emphasis in the similarities and distinctions of T cell dysfunctions in both kind of cancer (which must be resumed in a figure).
Indeed, the figures of the manuscript are quite simple or general, and does not show, the main messages of the review.
Specifics:
Figure 1: It requires to be better explained (what the lists in each box means?; which are protumoral or antitumoral signals?; are more important in solid or hematopoietic tumors?; etc.).
Line 81: "...T cells can become activated and start to proliferate." (and differentiate into effector and memory cells).
Line 87: Cytotoxic granules containing granzyme AND perforin.
Lines 110-111: despite that vaccines do not impact cancer treatment as ICI or CAR T cells do, nowadays there is a revival of cancer vaccines, particularly because they are needed for combine with ICI and our new knowledge of neoantigens. I suggest to include these new renascence of vaccines concept and may be do not be so categorical. Reference 13 is quite old, there are many current reviews on cancer vaccines that should be cited instead (including those that talk about neoantigen targeting vaccines).
Line 178: It could be a reflect of major tumor infiltration of CD8+ T cells in patients with immunogenic tumors, but yes, of course it requires more research.
Lines 202-204: I do not understand. I think there is a mistake here. Take a look.
Line 219: Explain CLL meaning
Line 266: As the authors are talking of all hematopoietic cancer types (in general), why then say "crosstalk of transformed B cells" (which is more specific)?
Line 359: where the excellent reviews on bystander cells are indicated?
Along the manuscript: use CD8+ T cells instead CD8 T cells
Figure 3: The legend must be sufficient for understanding the full message of the figure.
